# Impact of Extraskeletal Metastases on Skeletal-Related Events in Metastatic Castration-Resistant Prostate Cancer with Bone Metastases

**DOI:** 10.3390/cancers12082034

**Published:** 2020-07-24

**Authors:** Soraia Lobo-Martins, Arlindo R. Ferreira, André Mansinho, Sandra Casimiro, Kim Leitzel, Suhail Ali, Allan Lipton, Luís Costa

**Affiliations:** 1Oncology Division, Hospital de Santa Maria, 1649-035 Lisbon, Portugal; soraia.l.martins@chln.min-saude.pt (S.L.-M.); andr3.m@gmail.com (A.M.); 2Instituto de Medicina Molecular-João Lobo Antunes, Faculdade de Medicina, Universidade de Lisboa, 1649-028 Lisbon, Portugal; arlindo.ferreira@fundacaochampalimaud.pt (A.R.F.); scasimiro@medicina.ulisboa.pt (S.C.); 3Breast Unit, Champalimaud Clinical Center, Champalimaud Foundation, 1400-038 Lisbon, Portugal; 4Division of Hematology/Oncology, Penn State Health Milton S Hershey Medical Center, Hershey, PA17033, USA; kleitzel@pennstatehealth.psu.edu (K.L.); Suhail.Ali@va.gov (S.A.); alipton@pennstatehealth.psu.edu (A.L.)

**Keywords:** prostate cancer, metastatic castration-resistant prostate cancer, bone-targeted agents, bone metastases, visceral metastases

## Abstract

The therapeutic landscape of metastatic castration-resistant prostate cancer (mCRPC) has substantially evolved over the last decade. Nonetheless, a better understanding of bone-targeted agents (BTAs) action in mCRPC remains an unmet need. Theuse of BTAs aims to reduce the incidence of skeletal-related events (SREs) in patients with mCRPC. Less frequent BTA schedules are currently being studied to minimize adverse events. In this study, the impact of metastatic compartment (bone and extraskeletal metastases (BESM) vs. bone-only metastases (BOM)) on bone biomarker kinetics, time to first on-study SRE, and symptomatic skeletal events (SSEs) is evaluated. This is a retrospective analysis of the prospective, randomized, multicenter clinical trial of denosumab vs. zoledronic acid in patients with mCRPC and bone metastases. A total of 1901 patients were included, 1559 (82.0%) with BOM and 342 with BESM (18.0%). Bone metastases burden was balanced between groups. Baseline levels and normalization rates of corrected urinary N-terminal telopeptide and bone alkaline phosphatase did not differ between groups. However, BESM patients had a higher risk of SREs (adjusted HR 1.21; 95% CI 1.01–1.46; *p* = 0.043) and SSEs (adjusted HR 1.30; 95% CI 1.06–1.61; *p* = 0.014). This difference was more pronounced in the first 12 months of BTA treatment.In mCRPC, strategies of BTA schedule de-escalation may take into account presence of extraskeletal metastases.

## 1. Introduction

Prostate carcinoma is one of the most common cancers in men globally and the second cause of cancer death in this gender in Europe [1,2]. In these patients, bone is the most common site of metastatic disease (>90% of patients) [3] and bone metastases are frequently associated with detrimental bone outcomes—collectively referred to as skeletal-related events (SREs; pathological fracture, spinal cord compression, and radiotherapy or surgery to bone), which negatively impact quality of life and survival [3,4,5,6]. Many metastatic castration-resistant prostate cancer (mCRPC) patients also present metastatic disease outside the bone, with the type of metastatic spread (i.e., the metastatic compartment) having strong prognostic survival implications. In a meta-analysis with over 8000 mCRPC patients from nine phase III trials, men with visceral metastases had worse overall survival (OS)—13.5 months (95% confidence interval (CI) 12.7–14.4 months) for patients with liver metastases and 19.4 months (95% CI 17.8–20.7 months) for those with lung metastases compared with 20.8 months for patients with bone-only metastases (BOM) (95% CI20.8–21.9 months) [7].

Metastatic bone disease disrupts bone metabolism [8,9], weakening the structural integrity of bone and leading to skeletal-related events (SREs) [9]. Prostate cancer cells secrete factors, such as bone morphogenic proteins (BMPs), TGFβ, and endothelin-1 (ET-1), which increase osteoblastic activity and lead to osteoblastic metastases [10,11]. In this setting, bone-targeted agents (BTAs; denosumab and zoledronic acid (ZA)) reduce bone resorption and bone metastases-associated morbidity [1,4,5]. Bone biomarkers have been used to capture BTA action in bone, and both their baseline and post-BTA values are strongly prognostic [8,12,13,14,15].

While being generally well-tolerated drugs, in a patient population with increasingly longer survival, BTAs cumulative exposure has become a cause for concern, especially due to jaw osteonecrosis risk [3,16,17]. De-escalation strategies have been studied, aiming to provide the same efficacy as the standard schedule while improving patient adherence and safety. In fact, results from several studies have shown that the 12-weekly de-escalation regimen is noninferior to the 4-weekly dosage regimen in patients with bone metastases [17,18]. However, the use of the 12-week schedule might be troublesome in some patients, namely those with previous SREs and metastatic disease outside the bone [19]. Exploratory studies have shown that the type of metastatic spread (i.e., metastatic compartment as assessed by bone-only metastases (BOM) vs. bone plus extraskeletal metastases (BESM)) may impact SRE risk. In a cohort of breast cancer patients with bone metastases, patients with BESM displayed shorter interval until first SRE (HR 1.62, 95% CI 1.01–2.60, *p* = 0.0449) compared with patients with BOM [20], consistently high urinary N-terminal telopeptide (uNTX) levels, and an unpredictable pattern of bone marker variation over time, which was not normalized with BTAs [20]. Based on these findings, we hypothesized that patients with BESM may have increased bone marker levels and higher risk of SREs and symptomatic skeletal events (SSEs) than those with BOM.

In this large study of mCRPC with bone metastases, we investigated the impact of themetastatic compartment (BOM vs. BESM) onbone biomarker levels at baseline and after BTA introduction, time to first and subsequent on-study SREs/SSEs and OS.

## 2. Results

### 2.1. Cohort Description

A total of 1901 patients receiving either denosumab or ZA were identified for this study from the NCT00321620 trial, 1559 (82.0%) with BOM and 342 (18.0%) with BESM. Patient, disease, and treatment characteristics are summarized in Table 1. Compared to patients with BOM, patients with BESM were younger (≤65 years—31.5% vs. 40.4%, *p* = 0.013), had higher body mass index (BMI; median 27.7 (interquartile range (IQR) 25.3–30.5) vs. 26.9 (IQR 24.3–29.8) Kg/m^2^, *p* = 0.003), more frequently had node-positive disease at diagnosis (N1—74.0% vs. 0.0%, *p* < 0.001), and had higher total prostate-specific antigen (PSA) levels at diagnosis (≥10 ng/mL—88.6% vs. 83.9%, *p* = 0.029). More patients with BESM were under chemical castration (76.3% vs. 63.8%) compared withpatients with BOM. While on trial, more BESM patients received (26.6% vs. 11.1%, *p* < 0.001) or had previously received (39.8% vs. 19.9%, *p* < 0.001) chemotherapy or radiotherapy (39.2% vs. 30.2%, *p* = 0.001) compared with BOM patients.

### 2.2. Bone Disease-Specific Characteristics

Bone disease-specific characteristics were balanced between groups, namely time from cancer diagnosis to bone metastases (27.2 months in BESM vs. 23.9 months in BOM group, *p* = 0.784), bone metastases burden (>2 lesions—31.6% in BESM vs. 34.5% in BOM group, *p* = 0.300), and proportion of patients facing previous SREs (78.1% in BESM vs. 75.1% in BOM group, *p* = 0.248). Previous bisphosphonate use was similar between groups (no previous use—97.4% in BESM vs. 96.9% in BOM group, *p* = 0.660), as well as BTA type (ZA—52.9% in BESM vs. 49.0% in BOM group, denosumab—46.8% in BESM vs. 50.2% in BOM group, *p* = 0.291).

### 2.3. Bone Marker Variation

At baseline, mean corrected urinary N-telopeptide (uNTX) levels were 48 nmol BCE/mmol creatinine (IQR 27–100) in BOM and 48 nmol BCE/mmol creatinine(IQR 25–85) in BESM patients (*p* = 0.277) (Figure 1 and Appendix A). The proportion of patients with corrected uNTX normalization at 3 months was 80.2% (*n* = 364) and 85.6% (*n* = 77; *p* = 0.234), respectively (Figure 1A). Mean bone alkaline phosphatase (bALP) levels at baseline were 31 ng/mL (IQR 17–74) in BOM and 29 ng/mL (IQR 17–75) in BESM patients (*p* = 0.899). The proportion of patients with bALP normalization at 3 months was 29.0% (*n* = 236) in BOM and 32.9% (*n* = 56) in bone plus BESM patients (*p* = 0.310; Figure 1B).

### 2.4. Skeletal-Related Events

After a median follow-up of 20.1 months (IQR 15.9–23.8; balanced between arms), 27.9% (*n* = 530) of patients developed an on-study SSE and 38.2% (*n* = 727) an on-study SRE in the overall cohort (Table 2). The proportion of patients developing SREs was similar between groups (42.1% in BESM vs. 37.4% in BOM group). No substantial differences were found regarding SRE pattern, such as number of SRE and type of SRE (Table 2).

At 12 and 24 months, 64.1% (95% CI 61.3–66.8) vs. 55.9% (95% CI 49.4–61.8) and 44.2% (95% CI 40.3–48.0) vs. 35.8% (95% CI 27.4–44.3) of patients in BOM and BESM groups were free of SREs, respectively. The median time to an SRE was 14.9 months (IQR 5.6- not reached (NR)) for patients with BESM and 19.8 months (IQR 7.2–34.0) for patients with BOM. In univariate analysis, patients with BESM had a 25% higher risk of developing an SRE compared to patients with BOM (hazard ratio (HR) 1.25, 95% CI 1.04–1.51, *p* = 0.015) (Figure 2). Consistent results were found in multivariate analysis (HR 1.21, 95%CI 1.01–1.46, *p* = 0.043) and in multiple failure-time analysis (Table 3). In univariate analysis, features also associated with increased SRE risk included ECOG-PS 2 vs. ≤1 (HR 1.36, 95%CI1.01–1.82, *p* = 0.043); higher total PSA levels at diagnosis (≥10 ng/mL—HR 1.23, 95%CI1.02–1.50, *p* = 0.035); previous SREs (HR 1.45, 95%CI1.23–1.70, *p* < 0.001); previous (HR 1.29, 95%CI1.09–1.52, *p* = 0.003) or current (HR 1.40, 95%CI1.15–1.70, *p* = 0.001) chemotherapy treatment; and previous radiotherapy treatment (HR 1.26, 95%CI1.09–1.47, *p* = 0.002).

### 2.5. Symptomatic Skeletal Events

At 12 and 24 months, 74.7% (95% CI 72.1–77.1) vs. 64.2% (95% CI 57.8–69.8) and 58.0% (95% CI 54.1–61.6) vs. 50.1% (95% CI 41.9–57.7) of patients in BOM and BESM groups were free of SSEs, respectively. Median time to an SSE was 33.2 months (IQR 12.0-NR) for patients with BESM and 28.8 months (IQR 8.7-NR) for patients with BOM (Figure 3). In univariate analysis, patients with BESM had a 40% higher risk of developing an SSE compared with patients with BOM (HR 1.40, 95% CI 1.14–1.72, *p* = 0.002). Consistent results were found in multivariate analysis (HR 1.30, 95% CI 1.06–1.61, *p* = 0.014) and multiple failure-time analysis (Table 4). In univariate analysis, features also associated with a higher SSE risk included higher total PSA levels at diagnosis (≥10 ng/mL—HR 1.29, 95% CI 1.02–1.96, *p* = 0.031); previous SREs (HR 1.63, 95% CI 1.35–1.96, *p* < 0.001); previous (HR 1.32, 95% CI 1.09–1.60, *p* = 0.004) or current (HR 1.47, 95% CI 1.17–1.84, *p* = 0.001) chemotherapy; and previous radiotherapy (HR 1.48, 95% CI 1.24–1.76, *p* < 0.001).

### 2.6. Overall Survival

After a median follow-up of 16.6 months (IQR 10.7–23.9; balanced between arms), 49.2% (*n* = 935) of patients in the overall cohort died, 47.3% (*n* = 738) in the BOM and 57.6% (*n* = 197) in the BESM group (Appendix A). Median time to death was 15.8 months (IQR 8.2–27.9) for BESM and 20.5 months (IQR 9.8-NR) for BOM group. In univariate analysis, BESM patients had a 38% higher risk of death compared to BOM patients (HR 1.38, 95% CI 1.17–1.61; *p* < 0.001).

## 3. Discussion

The treatment of mCRPC has substantially evolved over the last decade, namely with the introduction of new therapeutic options. However, bone remains a major metastatic site in mCRPC, with important morbidity, quality of life, and survival implications. Assessment of a high-quality cohort derived from one of the largest clinical trials of mCRPC patients treated with BTAs is a major opportunity to improve knowledge and optimize BTA use in mCRPC.

Visceral metastases have long been considered a negative prognostic factor for survival in mCRPC [7,12,21]. In this large and high-quality cohort of patients with mCRPC and bone metastases receiving BTAs, patients with bone plus extraskeletal metastases had a similar baseline and 3-month normalization patterns of bone remodeling markers as patients with bone-only metastases, but a higher risk of SREs, SSEs, and death irrespective of bone disease volume and disease aggressiveness features. This difference was more evident in the first year after diagnosis of mCRPC with bone metastases.

Bone metastatic niche is a rich source of growth factors and other soluble molecules that are released by bone metabolism activation from mCRPC bone metastases [12,20,22]. Several of these biomarkers, such as uNTX and bALP, are amenable to serum or urine quantification and constitute strong predictors of survival and SRE risk in patients with bone metastases from several tumor types [8,12,20,22]. Indeed, a 2016 study by Lipton et al., using an integrated analysis of three similar phase III trials, showed that patients with bone turnover markers above the median after 3 months of antiresorptive therapy had significantly worse clinical outcomes, including OS and bone disease progression, compared to patients with bone turnover markers below the median [8].

In the present study, bone biomarker assessment, at baseline and 3 months after initiation of BTA therapy, was used to dissect the prognostic implications of different metastatic spread patterns (i.e., metastatic compartments) by analyzing patients with BOM and BESM. While a previous study of patients with breast cancer and bone metastases showed that BESM patients had a numerical trend for persistently higher uNTX levels and an erratic uNTX variation during ZA treatment [20], in the present cohort bone markers did not differ according to metastatic compartment both at baseline and at 3 months. While the nature of the present cohort provides a more definitive assessment on how bone biomarkers differ at baseline and vary over time, the short-term assessment (3 months) restricts the longitudinal understanding of bone marker dynamics at longer time intervals. In addition, despite the fact that all bone metastases activate bone metabolism, bone-forming vs. bone-degrading components are differently impacted in patients with bone metastases from breast and prostate cancer [23,24], a fact thatmay also contribute to explain the observed difference.

Despite similar bone biomarker levels at baseline and comparable short-term variation, patients with BESM had higher SRE and SSE risk compared with patients with BOM, and this seems to be particularly relevant within the first 12 months after diagnosis. The added SRE risk in patients with BESM was also identified by Tanaka and colleagues in a cohort of 534 breast cancer women who developed bone metastases [19]. The present study confirms that result in a large cohort of mCRPC patients, and taken together these findings challengethe current trend towards a reduced BTA scheduling frequency from every 4 weeks to every 12 weeks (especially for ZA) across all mCRPC patients [16,25,26]. Specifically, this study supports the hypothesis that the metastatic compartment has impact in bone outcomes and may be taken into account when considering de-escalation strategies to better tailor such approaches. Therefore, patients with BESM are at higher SRE risk and should receive a more conservative treatment schedule (i.e., every 4 weeks), at least during the first year of treatment. It remains unknown whether humoral factors affecting bone resorption are produced by extraskeletal metastases and/or whether cancer cells can circulate from different metastatic sites (outside of bone to bone and vice versa) to restimulate microenvironment [10,20,21]. Despite similar metastatic bone disease burden and disease biology, this cohort shows an impact of BESM on SREsthat is not captured by corrected uNTX and bALP levels. This may suggest that these biomarkers are not the best surrogates for such humoral factors. While the present study does not directly address these issues, it discloses new research avenues with potential therapeutic impact for optimizing bone outcomes in patients with mCRPC and raises relevant questions to understand cancer dynamics during advanced-stage of disease.

As anticipated, patients with BESM displayed worse survival outcomes. This is consistent with previous studies [21,27,28] and raises awareness of a patient population for whom novel approaches are urgently required.

Despite the high-quality dataand having included a large cohort with long-term follow-up, the present study has intrinsic limitations. It was a retrospective analysis, therefore subject to residual confounding, notwithstanding methodological accuracy. While all models were controlled for disease biology (Gleason score) and burden (bone metastases number), full disease biology is not captured by these features, whichmay partially explain the study findings.

## 4. Materials and Methods

### 4.1. Design and Study Population

This was a retrospective analysis of the prospective, randomized, multicenter registration clinical trial of denosumab vs. ZA in patients with mCRPC and bone metastases (NCT00321620) [4]. All participating patients were selected. Figure 4 details patient enrollment in the several preplanned analyses. Bone metastases diagnosis and extraskeletal metastases screening were conducted as prespecified per NCT00321620 protocol.

### 4.2. Study Aim and Outcomes

The main aim of this study was to investigate how metastatic compartment (BOM vs. BESM) impacts clinical outcomes and bone metabolism in patients with prostate cancer receiving BTAs. Specifically, the impact of metastatic compartment (BOM vs. BESM) on uNTX and bALP levels at baseline and normalization rates from baseline to 3 months was studies. We subsequently analyzed its impact on time to first on-study SRE/SSE and on OS. The 3-month-after-antiresorptive-treatment time point was selected to provide adequate therapy response time (no further time points were available). Outcomes were defined as per the study protocol [4].

### 4.3. Bone Marker Determination

In the NCT00321620 trial [4], urine specimens (from secondmorning void) and venous blood samples were collected to measure bone metabolism biochemical markers in all patients. Urinary bone marker to creatinine ratio was assessed for uNTX. Serum bone-specific alkaline phosphatase was also quantified. A central laboratory (Mayo Medical Laboratories, Rochester, MN, USA) performed urine and serum biochemical bone metabolism marker measurements for all patients. Normal uNTX cut-off was <64 nmol BCE/mmol creatinine and bALP cut-off was <22 ng/mL [13].

### 4.4. Statistical Analysis

Descriptive statistics of baseline clinical, pathological, and treatment characteristics were performed for the overall cohort and for BOM and BESM subgroups. Univariate comparisons were performed using a chi-square or *t*-test, as appropriate.

uNTX and bALP variation from baseline to 3 months according to metastatic compartment was tested using paired *t*-test. Time-to-event outcomes, survival, and cumulative incidence were plotted using the Kaplan–Meier method. Univariate and multivariate differences between survival rates according to metastatic compartment were tested using standard univariate and multivariate Cox proportional hazards model and the Andersen and Gill model for multiple failure time data. Multivariate analyses were corrected for age, ECOG-PS, Gleason score, total PSA at diagnosis, number of bone lesions, previous SREs, castration type, and previous surgery or radiotherapy. Given the strong association between visceral involvement and chemotherapy, no adjustment was performed for previous or current chemotherapy treatment.

All tests were two-sided with a 95% confidence interval (CI), and *p*-value < 0.05 was considered statistically significant. Statistical analyses were conducted using Stata 13.1 (https://www.stata.com/, StataCorp LP, College Station, TX, USA).

### 4.5. Ethical Statement

Approval from appropriate research Ethics Committees was obtained from each study center. All patients provided written informed consent before any study-specific procedure.

## 5. Conclusions

In this large mCRPC registration and high-quality study, despite similar bone marker levels at baseline and at 3 months, patients with bone metastases plus extraskeletal metastases had higher risk of SRE and SSE than patients with BOM. Given the higher risk for adverse outcomes, strategies of BTA schedule de-escalation should consider the impact of metastatic compartment, particularly during the first treatment year.

## Figures and Tables

**Figure 1 cancers-12-02034-f001:**
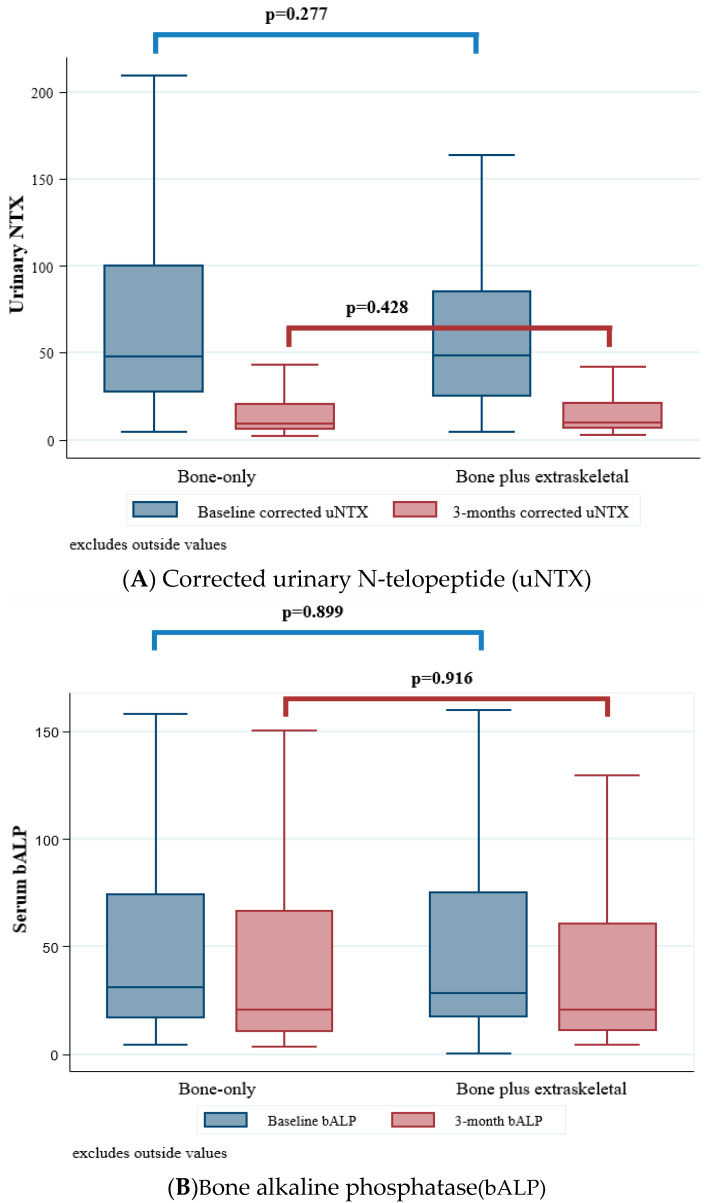
Bone markers at baseline and week 13. (**A**) Corrected uNTX at baseline and week 13; (**B**) bALP at baseline and week 13.bALP, bone alkaline phosphatase; uNTX, urinary N-terminal telopeptide.

**Figure 2 cancers-12-02034-f002:**
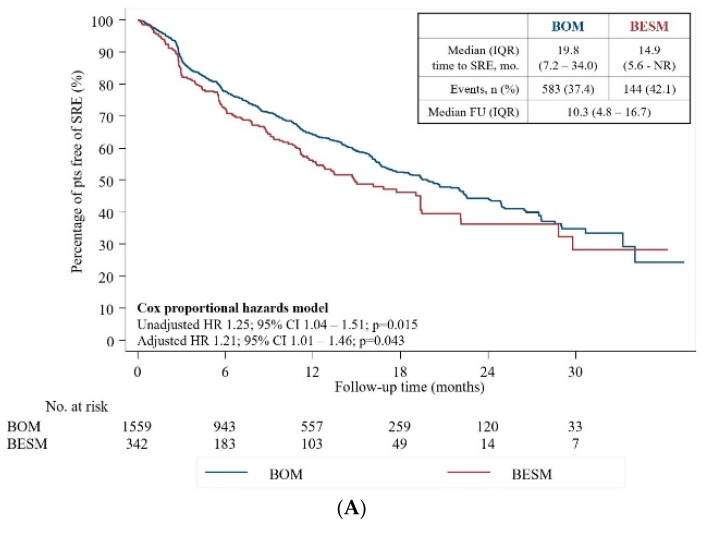
(**A**) Time-to-first on-study SRE; (**B**) Hazard for first on-study SRE.BESM, Bone and extraskeletal metastases; BOM, bone-only metastases; CI, confidence interval; ESD, extraskeletal disease; HR, hazard ratio; IQR, interquartile range; NR, not reached; pts, patients; SRE, skeletal-related event.

**Figure 3 cancers-12-02034-f003:**
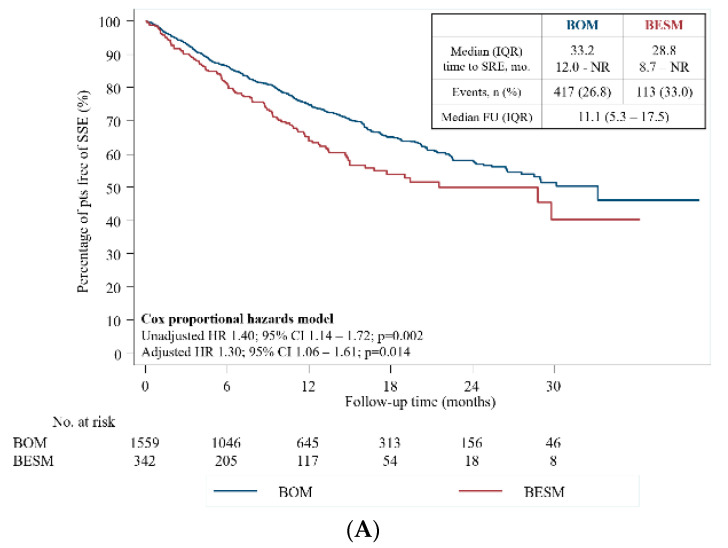
(**A**) Time-to-first on-study SSE; (**B**) Hazard for first on-study SSE. Hazards model.BESM, Bone and extraskeletal metastases; BOM, bone-only metastases; CI, confidence interval; ESD, extraskeletal disease; HR, hazard ratio; IQR, interquartile range; NR, not reached; pts, patients; SSE, symptomatic skeletal event.

**Figure 4 cancers-12-02034-f004:**
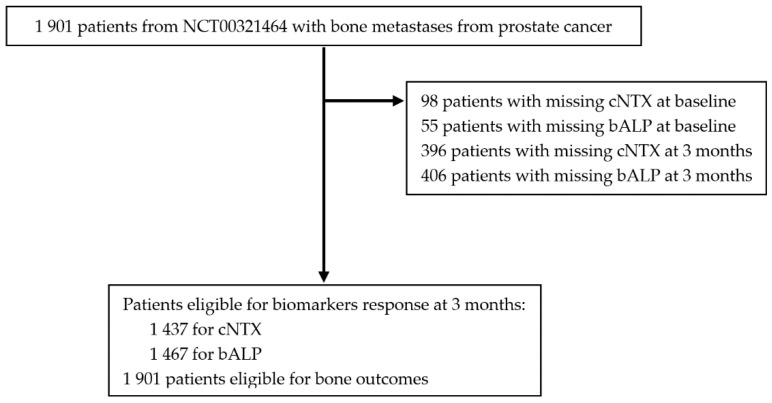
Consort study flowchart. bALP, bone alkaline phosphatase; BPI, Brief Pain Score; cNTX, corrected N-terminal telopeptide.

**Table 1 cancers-12-02034-t001:** Patient demographic/clinicopathological characteristics and type of concomitant treatment according to metastatic compartment.

Characteristic	Overall Cohort	BOM	BESM	*p*-Value (BOM vs. BESM)
Number of patients, *n* (%)	1901 (100)	1559 (82.0)	342 (18.0)	-
**Demographic and clinicopathological characteristics**				
Age, years				
Median P25–P75 Range	74 65–76 35–96	74 65–76 35–96	66 64–75 45–86	<0.001
Age (years), *n* (%)				
≤50 >50 to ≤65 >65 to ≤75 >75	14 (0.7) 615 (32.4) 726 (38.2) 546 (28.7)	10 (0.6) 481 (30.9) 605 (38.8) 463 (29.7)	4 (1.2) 134 (39.2) 121 (35.4) 83 (24.3)	0.013
ECOG performance status, *n* (%)				
≤1 2	1768 (93.0) 133 (7.0)	1448 (92.9) 111 (7.1)	320 (93.6) 22 (6.4)	0.652
Body mass index				
Median IQR Missing, *n* (%)	27.0 24.5–30.0 47 (2.5)	26.9 24.3–29.8 39 (2.5)	27.7 25.3–30.5 8 (2.3)	0.003
T stage at disease diagnosis, *n* (%)				
T1-T2 T3 T4 Missing	725 (44.8) 693 (42.8) 200 (12.4) 283 (14.9)	604 (45.8) 559 (42.4) 156 (11.8) 240 (15.4)	121 (40.5) 134 (44.8) 44 (14.7) 43 (12.6)	0.172
N stage at disease diagnosis, *n* (%)				
N0 N1	1648 (86.7) 253 (13.3)	1559 (100) 0	89 (26.0) 253 (74.0)	<0.001
M stage at disease diagnosis, *n* (%)				
M0 M1 Missing	832 (55.7) 662 (44.3) 407 (21.4)	685 (55.9) 541 (44.1) 333 (21.4)	147 (54.9) 121 (45.1) 74 (21.6)	0.760
Gleason score, *n* (%)				
≤6 7 ≥8	355 (18.7) 744 (39.1) 802 (42.2)	297 (19.0) 614 (39.4) 648 (41.6)	58 (17.0) 130 (38.0) 154 (45.0)	0.452
PSA at disease diagnosis, ng/mL				
Median IQR	60 19–213	57 18–207	78 23–247	0.021
PSA at disease diagnosis, *n* (%)				
<10 ng/mL ≥10 ng/mL	290 (15.3) 1611 (84.7)	251 (16.1) 1308 (83.9)	39 (11.4) 303 (88.6)	0.029
Time from cancer diagnosis to bone metastases, months				
Median IQR	24.5 1.8–68.6	23.9 2.0–69.3	27.2 1.2–66.4	0.784
Type of bone metastases, *n* (%)				
Lytic Blastic Mixed Unknown/Not seen	71 (4.8) 1138 (76.5) 278 (18.7) 414 (21.8)	57 (4.6) 961 (77.6) 221 (17.8) 320 (20.5)	14 (5.6) 177 (71.4) 57 (23.0) 94 (27.5)	0.109
Number of bone metastases, *n* (%)				
≤2 >2 to ≤4 >4	1255 (66.0) 329 (17.3) 317 (16.7)	1021 (65.5) 271 (17.4) 267 (17.1)	234 (68.4) 58 (17.0) 50 (14.6)	0.484
Type of visceral metastases, *n* (%) ^1^				
Liver Lung Other sites	-	-	36 (10.5) 58 (17.0) 294 (86.0)	-
TPrevious SRE, *n* (%)				
Yes No	463 (24.4) 1438 (75.6)	388 (24.9) 1171 (75.1)	75 (21.9) 267 (78.1)	0.248
**Treatment characteristics**				
BTA type, *n* (%)				
Zoledronic acid Denosumab No BTA	943 (49.6) 945 (49.7) 13 (0.7)	764 (49.0) 783 (50.2) 12 (0.8)	181 (52.9) 160 (46.8) 1 (0.3)	0.291
Previous bisphosphonate treatment, *n* (%)				
Yes No	57 (3.0) 1844 (97.0)	48 (3.1) 1511 (96.9)	9 (2.6) 333 (97.4)	0.660
Castration type, *n* (%)				
Castration type, *n* (%) Chemical Surgical Both	1256 (66.1) 92 (4.8) 553 (29.1)	995 (63.8) 85 (5.5) 479 (30.7)	261 (76.3) 7 (2.1) 74 (21.6)	<0.001
Currently receiving CT, *n* (%)				
Yes No	264 (13.9) 1637 (86.1)	173 (11.1) 1386 (88.9)	91 (26.6) 251 (73.4)	<0.001
Previous CT, *n* (%)				
Yes No	446 (23.5) 1455 (76.5)	310 (19.9) 1249 (80.1)	136 (39.8) 206 (60.2)	<0.001
Previous RT, *n* (%)				
Yes No	604 (31.8) 1297 (68.2)	470 (30.2) 1089 (69.8)	134 (39.2) 208 (60.8)	0.001
Previous prostatectomy, *n* (%)				
Yes No	313 (16.5) 1588 (83.5)	247 (15.8) 1312 (84.2)	66 (19.3) 276 (80.7)	0.119
Previous antineoplastic surgery (any), *n* (%)				
Yes No	832 (43.8) 1069 (56.2)	655 (42.0) 904 (58.0)	177 (51.8) 165 (48.3)	0.001

BOM, bone-only metastases; BTA, bone-targeted agent; CT, chemotherapy; ECOG, Eastern Cooperative Oncology Group; BESM, bone and extraskeletal metastases; IQR, interquartile range; PSA, prostate-specific antigen; RT, radiotherapy; SRE, skeletal-related event. ^1^ Patients could present with metastases in several visceral sites. As missing data arenot considered for the nonmissing proportion of patients, proportion sum may exceed 100% in cases of missing data.

**Table 2 cancers-12-02034-t002:** Occurrence and type of SREs according to metastatic compartment.

Characteristic	BOM	BESM	*p*-Value
Development of any on-study SRE, *n* (%)			
Yes No	583 (37.4) 976 (62.6)	144 (42.1) 198 (57.9)	0.105
Development of symptomatic on-study SREs, *n* (%)			
Yes No	417 (26.8) 1142 (73.3)	113 (33.0) 229 (67.0)	0.019
Number of SREs			
Median (IQR) Min.–max.	0 0–1	0 0–1	0.099
Number of SREs, *n* (%)			
0 1 2 ≥3	976 (62.6) 383 (24.6) 127 (8.1) 73 (4.7)	198 (57.9) 91 (26.6) 38 (11.1) 15 (4.4)	0.224
Pts with radiotherapy to bone as first SRE, *n* (%) ^1^			
Yes No	309 (53.0) 274 (47.0)	78 (54.2) 66 (45.8)	0.802
Pts with pathological fracture as first SRE, *n* (%) ^1^			
Yes No	233 (40.0) 350 (60.0)	55 (38.2) 89 (61.8)	0.697

BOM, bone-only metastases; BESM, bone and extraskeletal metastases; IQR, interquartile range; pts, patients; SRE, skeletal-related events. ^1^ Only patients with on-study SREs.

**Table 3 cancers-12-02034-t003:** Risk for development of first on-treatment SRE.

Characteristic	Risk for SRE Development		
	HR	95% CI	*p*-Value
**Univariate analysis**			
Metastatic compartment			
Bone-only Bone + extraskeletal metastases	Reference 1.25	Reference 1.04–1.51	0.015
Age (years)			
≤50 >50 to ≤65 >65 to ≤75 >75	Reference 0.47 0.38 0.43	Reference 0.26–0.85 0.21–0.70 0.23–0.79	0.014 0.002 0.006
ECOG performance status			
≤1 2	Reference 1.36	Reference 1.01–1.82	0.043
Gleason score			
≤6 7 ≥8	Reference 0.97 1.19	Reference 0.79–1.20 0.98–1.46	0.795 0.977
PSA at disease diagnosis, ng/mL			
<10 ≥10	Reference 1.23	Reference 1.02–1.50	0.035
Type of bone metastases			
Lytic Blastic Mixed	Reference 0.92 0.93	Reference 0.63–1.33 0.62–1.41	0.648 0.738
Number of bone metastases			
≤2 >2	Reference 1.12	Reference 0.96–1.30	0.162
Previous SREs			
No Yes	Reference 1.45	Reference 1.23–1.70	<0.001
Castration type			
Chemical Surgical Both	Reference 0.49 0.78	Reference 0.33–0.75 0.66–0.93	0.001 0.004
Currently receiving chemotherapy			
No Yes	Reference 1.40	Reference 1.15–1.70	0.001
Previous chemotherapy			
No Yes	Reference 1.29	Reference 1.09–1.52	0.003
Previous radiotherapy			
No Yes	Reference 1.26	Reference 1.09–1.47	0.002
Previous anti-neoplastic surgery (any)			
No Yes	Reference 0.96	Reference 0.83–1.11	0.600
**Multivariate model 1; *n* = 1901**			
**Covariates:** Age, ECOG PS, Gleason score, PSA at diagnosis, number of bone lesions, previous SREs, castration type, and previous surgery and radiotherapy.			
Metastatic compartment			
Bone-only Bone + extraskeletal metastases	Reference 1.21	Reference 1.01–1.46	0.043
**Andersen-Gill model for multiple failure-time data**			
Unadjusted HR 1.28; 95% CI 1.04–1.58; *p* = 0.018; Adjusted HR 1.23; 95% CI 1.00–1.52; *p* = 0.048			

CI, confidence interval; ECOG PS, Eastern Cooperative Oncology Group performance status;HR, hazard ratio; PSA, prostate-specific antigen; SRE, skeletal-related event.

**Table 4 cancers-12-02034-t004:** Risk for development of first on-treatment SSE.

Characteristic	Risk for SSE Development		
	HR	95% CI	*p*-Value
**Univariate analysis**			
Metastatic compartment			
Bone-only Bone + extra-bone metastases	Reference 1.40	Reference 1.14–1.72	0.002
Age (years)			
≤50 >50 to ≤65 >65 to ≤75 >75	Reference 0.53 0.38 0.34	Reference 0.27–1.03 0.19–0.73 0.17–0.68	0.063 0.004 0.002
ECOG performance status			
≤1 2	Reference 1.39	Reference 0.99–1.95	0.055
Gleason score			
≤6 7 ≥8	Reference 0.92 1.27	Reference 0.72–1.19 1.01–1.62	1.188 1.619
PSA at disease diagnosis, ng/mL			
<10 ≥10	Reference 1.29	Reference 1.02–1.63	0.031
Type of bone metastases			
Lytic Blastic Mixed	Reference 0.99 0.90	Reference 0.63–1.55 0.55–1.48	0.952 0.672
Number of bone metastases			
≤2 >2	Reference 1.07	Reference 0.89–1.28	0.454
Previous SREs			
No Yes	Reference 1.63	Reference 1.35–1.96	<0.001
Castration type			
Chemical Surgical Both	Reference 0.37 0.67	Reference 0.21–0.64 0.54–0.82	<0.001 <0.001
Currently receiving chemotherapy			
No Yes	Reference 1.47	Reference 1.17–1.84	0.001
Previous chemotherapy			
No Yes	Reference 1.32	Reference 1.09–1.60	0.004
Previous radiotherapy			
No Yes	Reference 1.48	Reference 1.24–1.76	<0.001
Previous antineoplastic surgery (any)			
No Yes	Reference 0.93	Reference 0.79–1.11	0.439
**Multivariate model; *n* = 1901**			
**Covariates:** Age, ECOG PS, Gleason score, PSA at diagnosis, number of bone lesions, previous SREs, castration type, and previous surgery and radiotherapy			
Metastatic compartment			
Bone-only Bone + extraskeletal metastases	Reference 1.30	Reference 1.06–1.61	0.014

CI, confidence interval; ECOG PS, Eastern Cooperative Oncology Group performance status; HR, hazard ratio; PSA, prostate-specific antigen; SRE, skeletal-related event; SSE, symptomatic skeletal event.

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
