# Peer review of "Impact of Extraskeletal Metastases on Skeletal-Related Events in Metastatic Castration-Resistant Prostate Cancer with Bone Metastases"

_cancers, 2020, doi:10.3390/cancers12082034_

Round 1
Reviewer 1 Report
The paper reported a retrospective analysis on mCRPC patients enrolled in a randomized study comparing denosumab and zoledronic acid. This study was completed in 2012 and this significantly limited the relevance of the study results. Less than 40% were receiving or have previously received chemotherapy and ARTA had not yet been developed. Today all mCRPC pts receive active treatments which are able not only to prolong the OS but also to significantly reduce SRE. Thus the relevance of the paper conclusions are of little value in the actual therapeutic landscape of mCRPC
Author Response
Reviewer 1
- Comment: "The paper reported a retrospective analysis on mCRPC patients enrolled in a randomized study comparing denosumab and zoledronic acid. This study was completed in 2012 and this significantly limited the relevance of the study results. Less than 40% were receiving or have previously received chemotherapy and ARTA had not yet been developed. Today all mCRPC pts receive active treatments which are able not only to prolong the OS but also to significantly reduce SRE. Thus the relevance of the paper conclusions are of little value in the actual therapeutic landscape of mCRPC.”
Author reply: We acknowledge that the therapeutic landscape of mCRPC has substantially evolved over the last decade, namely with the introduction of new therapeutic options. Nonetheless, bone remains a major metastatic disease site in mCRPC, with important morbidity, quality of life, and survival implications. In our opinion, discussing bone-targeted agents (BTAs) is, even now, a matter of major importance. Some therapeutic advances came from incremental gains in management of previous therapies and we believe that a better dissection of BTA benefit in mCRPC remains an unmet need. In addition, while referring to a study completed in 2012, this is one of the largest clinical trials with mCRPC patients treated with BTAs. Furthermore, this is a registration clinical trial with high-quality data, reason why we consider it to be the best source of information in the subject.
We further acknowledge that recent treatment innovations impacted the risk for developing SREs and other unfavorable cancer outcomes. That said, ERA 223 trial, in which addition of abiraterone acetate and prednisone to Radium-223 in patients with mCRPC was studied, is an example of the central role of BTAs. While the use of bisphosphonates or denosumab was permitted if patients were receiving them at baseline, BTAs could not be initiated during the study due to potential confounding effect on primary outcome (symptomatic SRE-free survival). In such study, patients treated with the combination of bisphosphonate/denosumab plus Radium-223 had a longer symptomatic skeletal event-free survival, as well as lower fracture rate and numerically longer median overall survival. This example emphasizes the role of zoledronic acid/denosumab on bone outcomes even in presence of innovative approaches. Radium-223 early access program (Phase IIIb) subanalysis is consistent with this data, as the addition of denosumab to Radium-223 resulted in a longer OS and time to SSE.
From another perspective, the impact of metastatic compartment on bone outcomes has not been tested in recent abiraterone or enzalutamide clinical trials, highlighting the need to optimize bone outcomes using bone-directed agents.
Reviewer 2 Report
The paper by Soraia Lobo-Martins, et al. entitled “Impact of extraskeletal disease on skeletal-related events in metastatic castration-resistant prostate cancer with bone metastases – Analysis from a phase III clinical trial” is a comparative analysis between castration-resistant prostate cancer patients with bone-only metastasis and patients with bone and extra-skeletal metastasis. This is a retrospective analysis of a large-scale randomized clinical trial (NCT00321620, Lancet. 2011; 377(9768): 813-22.). The authors found that patients with bone and extra-skeletal metastasis had a higher risk of skeletal-related events and symptomatic skeletal events. Also, that difference was more pronounced in the first 12 months of bone-targeted agents treatment. Based on the results, the authors proposed that in metastatic castration-resistant prostate cancer, strategies of bone-targeted agents schedule de-escalation should consider presence of extra-skeletal metastasis, particularly during the first treatment year. The study is well conducted and the results seem reliable. The data is presented clearly and is presented in a scientifically sound manner. These findings will be of interest to readers of Cancers, however, I have the following concerns.
Comments
1. The key point of this paper is a comparison between patients with bone-only metastasis and patients with bone and extra-skeletal metastasis. A brief description of how the authors detected bone metastasis and performed screening extra-skeletal metastasis would be important, although the authors cited the original paper of the clinical trial.
2. In Figures, the vertical axis labels, skeletal-related events and symptomatic skeletal events would be free of skeletal-related events and free of symptomatic skeletal events, respectively.
3. The results are presented clearly and are reliable, however, it would be an overstatement to mention that strategies of bone-targeted agents schedule de-escalation should consider presence of extra-skeletal metastasis because this analysis is not focused on the effect of de-escalation. That would be a misleading expression toward readers.
4. No ethical statements are seen on the current form of the manuscript.
Author Response
Reviewer 2
- Comment: “1. The key point of this paper is a comparison between patients with bone-only metastasis and patients with bone and extraskeletal metastasis. A brief description of how the authors detected bone metastasis and performed screening extraskeletal metastasis would be important, although the authors cited the original paper of the clinical trial.”
Author reply: Bone metastases diagnosis and extraskeletal metastases screening were performed as pre-specified per NCT00321620 protocol (Fizazi et al, Lancet 2011). Although protocol was not co-published, this is a high-quality registration clinical trial and for this ancillary study we used the same main study outcomes.
In section 4.1 Design and study population, one should now read “This was a retrospective analysis of the prospective, randomized, multicenter registration clinical trial of denosumab vs ZA in patients with mCRPC and bone metastases (NCT00321620) [4]. All participating patients were selected. Figure 4 details patient enrollment in the several pre-planned analyses. Bone metastases diagnosis and extraskeletal metastases screening were conducted as pre-specified per NCT00321620 protocol.”
- Comment: “2. In Figures, the vertical axis labels, skeletal-related events and symptomatic skeletal events would be free of skeletal-related events and free of symptomatic skeletal events, respectively.”
Author reply: Thank you for noticing this aspect. According to the reviewer’s comment, vertical axis now refers to the proportion of patients free of SRE and SSE.
(see Figures in attached Editor reply document)
(A)
(B)
Figure 2. (A) Time-to-first on-study SRE; (B) Hazard for first on-study SRE.
BESM, Bone and extraskeletal metastases; BOM, bone-only metastases; CI, confidence interval; HR, hazard ratio; IQR, interquartile range; NR, not reached; pts, patients; SRE, skeletal-related event.
(see Figures in attached Editor reply document)
(A)
(B)
Figure 3. (A) Time-to-first on-study SSE; (B) Hazard for first on-study SSE. Hazards model.
BESM, Bone and extraskeletal metastases; BOM, bone-only metastases; CI, confidence interval; HR, hazard ratio; IQR, interquartile range; NR, not reached; pts, patients; SSE, symptomatic skeletal event.
- Comment: “3. The results are presented clearly and are reliable, however, it would be an overstatement to mention that strategies of bone-targeted agents schedule de-escalation should consider presence of extraskeletal metastasis because this analysis is not focused on the effect of de-escalation. That would be a misleading expression toward readers.”
Author reply: We acknowledge that the considered affirmation could be an overstatement and have rephrased it both in the abstract and in the discussion sections. In the abstract, we replaced “In mCRPC, strategies of BTA schedule de-escalation should consider presence of ESD, particularly during the first treatment year” for “In mCRPC, strategies of BTA schedule de-escalation may take into account presence of BESM.” In section 3. Discussion, we replaced “Specifically, this study supports the concept that some patient subgroups may not be optimal candidates for de-escalation and require a more tailored approach” for “Specifically, this study supports our thesis that the metastatic compartment has impact in bone outcomes and should be taken into account when considering de-escalation strategies to better tailor such approaches.” Also, in section 5. Conclusions, we replaced “In light of these results, strategies to de-escalate BTAs in the context of mCRPC should take into account presence of ESD and privilege an every-4-week schedule of administration, particularly during the first treatment year” for “In light of these results, strategies of BTA schedule de-escalation should consider the impact of metastatic compartment, particularly during the first treatment year.”
- Comment: “4. No ethical statements are seen on the current form of the manuscript.”
Author reply: Thank you for signaling this issue. We have now introduced a new section regarding ethical statements.
In Methods section, in the new section “4.5 Ethical statement”, we have added: “Approval from appropriate research ethics committees was obtained from each study center. All patients provided written informed consent before any study-specific procedure.”

Reviewer 3 Report
Manuscript Cancers- 819508
Reviewer comments
The manuscript entitled " Impact of extraskeletal disease on skeletal-related events in metastatic castration-resistant prostate cancer with bone metastases – Analysis from a phase III clinical trial " by Soraia Lobo-Martins, Arlindo Ferreira et al., presents original data on the retrospective analyses of a phase III clinical trial focusing on the consequences of the presence of extraskeletal metastases in addition to bone metastases of metastatic castration-resistant prostate cancer (mCRPC) on the occurrence of skeletal-related events (SRE) and symptomatic skeletal events (SSE).
Presented results are very interesting but the reviewer has several comments (major and minor) that need to be addressed to definitively convince the reviewer.
Major comments
First comment is related to the used terminology. All patients integrated in this study have bone metastases and some of them also extra-skeletal (visceral) metastases so forming two groups called by Authors: Bone Only Disease (BOD) and Extra-Skeletal Disease (ESD). The use of the word “Disease” is somehow perturbing. “Metastases” has to be used! Moreover ESD suggests that nothing occurs in bone what is not the case as all patients have bone metastases. So the terminology has to be changed, for example: Bone Only Metastases group (BOM) and Bone and Extra-Skeletal Metastases group (BESM).
Patients enrolled in the clinical trial have received two different inhibitors of bone resorption: Zoledronate and Denosumab. These two drugs also have other biological actions, for instance on the immune system or directly on the metastatic cell, with specificities for each of them. The distribution of the two treatments in the two groups of patients is not significantly different (Table 1) but this does not mean that the two drugs have the same effect in each group? Have authors analyzed the potential difference of SRE occurrence in the Bone and Extra-Skeletal Metastases group with the two drugs? This is of major importance for the whole interpretation of the data.
The median age variation between the two groups of patients and the apparent relationship with expression level of PSA is questioning! Have relationship already being described on the one hand between PSA expression level and age of patients and on the other hand between PSA and multi-sites metastases occurrence? Moreover is PSA expression level in the tumor cell similar in the prostate site and the different metastatic sites? Such questions have to be discussed in the manuscript.
Minor comments
In the Table 1, in Type of visceral metastases, n (%), the addition of the different sites is not 100 %?
Supplemental Table 1 is not supplemental as present in the text. Renumbering of the Tables is necessary.
In supplemental Table1 3-months is used and 13-weeks in the Figure1. Harmonization required if considering the same age!
Line 128, the sentence is not clear! “In contrast” to WHAT?
The Reviewer will appreciate the comments to be taken into account in a revised version of the manuscript.
Author Response
Reviewer 3
- Comment: “The manuscript entitled " Impact of extraskeletal disease on skeletal-related events in metastatic castration-resistant prostate cancer with bone metastases – Analysis from a phase III clinical trial " by Soraia Lobo-Martins, Arlindo Ferreira et al., presents original data on the retrospective analyses of a phase III clinical trial focusing on the consequences of the presence of extraskeletal metastases in addition to bone metastases of metastatic castration-resistant prostate cancer (mCRPC) on the occurrence of skeletal-related events (SRE) and symptomatic skeletal events (SSE). Presented results are very interesting but the reviewer has several comments (major and minor) that need to be addressed to definitively convince the reviewer.”
Author reply: We thank you for this comment. No changes were performed.
- Comment: “First comment is related to the used terminology. All patients integrated in this study have bone metastases and some of them also extraskeletal (visceral) metastases so forming two groups called by Authors: Bone Only Disease (BOD) and Extraskeletal Disease (ESD). The use of the word “Disease” is somehow perturbing. “Metastases” has to be used! Moreover, ESD suggests that nothing occurs in bone what is not the case as all patients have bone metastases. So the terminology has to be changed, for example: Bone Only Metastases group (BOM) and Bone and Extraskeletal Metastases group (BESM).”
Author reply: We appreciate and kindly accept the suggestion for nomenclature change, which has now been adopted in the revised manuscript version: BOD (bone-only disease) and ESD (extraskeletal disease) were replaced by BOM (bone-only metastases) and BESM (bone and extraskeletal metastases).
- Comment: “Patients enrolled in the clinical trial have received two different inhibitors of bone resorption: Zoledronate and Denosumab. These two drugs also have other biological actions, for instance on the immune system or directly on the metastatic cell, with specificities for each of them. The distribution of the two treatments in the two groups of patients is not significantly different (Table 1) but this does not mean that the two drugs have the same effect in each group? Have authors analyzed the potential difference of SRE occurrence in the Bone and Extraskeletal Metastases group with the two drugs? This is of major importance for the whole interpretation of the data.”
Author reply: We acknowledge that both zoledronic acid and denosumab may have systemic effects extending beyond their bone remodeling effect. However, as referred by the reviewer, the use of denosumab and zoledronic acid is well balanced in the two arms, thus it is very unlikely that it may have impacted study results. Moreover, this was not part of the initial study protocol, therefore such analysis is out of the scope of the current project, as agreed with Amgen.
- Comment: “The median age variation between the two groups of patients and the apparent relationship with expression level of PSA is questioning! Have relationship already being described on the one hand between PSA expression level and age of patients and on the other hand between PSA and multi-sites metastases occurrence? Moreover is PSA expression level in the tumor cell similar in the prostate site and the different metastatic sites? Such questions have to be discussed in the manuscript.”
Author reply: We appreciate this comment. However, no relationship between age and PSA levels was stated in the original manuscript, either in the text or on tables or figures. Therefore, no changes were included in this revised manuscript version.
- Comment: “In Table 1, in Type of visceral metastases, n (%), the addition of the different sites is not 100 %?”
Author reply:
We thank the reviewer for noting this error, now corrected. This issue occurred due to data transcription from a previous table while drafting the final manuscript tables. All manuscript tables have now been revised and no additional errors were detected.
- Comment: “Supplemental Table 1 is not supplemental as present in the text. Renumbering of the Tables is necessary.”
Author reply: Supplemental Table 1 has been reallocated to Appendix A.
- Comment: “In supplemental Table1 3-months is used and 13-weeks in the Figure1. Harmonization required if considering the same age!”
Author reply: Figure 1 has been corrected to 3-months, as displayed in supplemental Table 1.
(see Figures in attached Editor reply document)
(A) Corrected uNTX
(A) Corrected bALP
- Comment: “Line 128, the sentence is not clear! “In contrast” to WHAT?”
Author reply: We kindly accept your correction. “In contrast” was removed. Accordingly, it can be read “(…) (Table 2). The proportion of patients developing SREs was similar between groups (42.1% in ESM vs. 37.4% in BOM).”
The authors hope that these revisions have adequately addressed the associate editor and reviewers’ questions. Notwithstanding, we remain available for additional questions. Thank you for your attention.

Reviewer 4 Report
Lobo-Martins et al. present here a new study based on the analysis of data available in the RCT Zol vs Dmab in metastatic castration resistant prostate cancer with bone metastases (Fizazin Lancet 2011). They split available patients into 2 groups: bone only metastases (BOM) vs bone and extraskeletal metastases (BESM). From the 1901 patients, they included 1559 (82%) BOM and 342 (18%) BESM. Most of the characteristics remained balanced in the two groups concerning the cancer medical history (ECOG, time from diagnostic, size of primary tumor) and bone metastasis presentation (SRE history at inclusion, bone metastasis burden, uNTX level and correction at 3 months, previous bisphosphonate treatment) and treatment allocation (Zol vs Dmab). Nevertheless, it is probably not the same tumor agressivity profile and the same patient profile. In BESM group, patients are younger, with higher PSA, higher N score, different castration method, and received more chemotherapy. It is the main limitation of this study since it introduces uncontrolled biaised (eg: steroids given in association with chemotherapy ? bone toxicity of chemotherapy? Sarcopenia ? risk of fall ? ...). The authors found a higher rate of symptomatic SRE in BESM patients at 12 and 24 months and SRE occurred earlier during follow-up.
In fact, the authors identified a more severe and aggressive subpopulation of mCRBC with a poorer overall survival and a higher risk of SRE. I suggest that this should be more clearly mentioned in the conclusion (abstract, first sentence of discussion and conclusion) and the limitation should be acknowledged in the discussion.
Otherwise, the paper is well written. Graph and tables are fine.
Comment
- I could not judge the format of the paper since I received a pdf with all the tracking changes for peer review.
- What is the reason to finish with methods in an epidemiological study (IMRAD)? Please replace it at its regular location.
- I suggest a more explicit title with no mention of phase III since they lost the structure of the trial. “Bone metastatic patients with extra-skeletal from mCRPC have a higher risk of SRE independently of bone burden and bone turnover markers”
- I suggest to underline in the discussion that this BESM population from mCRPC is particularly at risk for symptomatic SRE and physicians should pay a particular attention to prevent them.
Round 2
Reviewer 1 Report
The authors reply did not change my feelings concerning the article
Author Response
The authors acknowledge this comment. The following changes have been made to the article accordingly:
Abstract (line 19): “The therapeutic landscape of mCRPC has substantially evolved over the last decade. Nonetheless, a better understanding of BTA benefit in mCRPC remains an unmet need. Use of bone-targeted agents (BTA) ….”
Discussion (line 197): “mCRPC treatment has substantially evolved over the last decade, namely with the introduction of new therapeutic options. However, bone remains a major metastatic site in mCRPC, with important morbidity, quality of life, and survival implications. Assessment of a high-quality cohort derived from one of the largest clinical trials of BTA-treated mCRPC patients is a major opportunity to improve knowledge and optimize BTA use in mCRPC.”
Reviewer 3 Report
Authors have answered all comments and explained why when unable to respond.
The Reviewer has no more comment.
Author Response
The authors appreciate this comment. No changes have been made.